# Frequent exchange of the DNA polymerase during bacterial chromosome replication

**Thomas R Beattie[1†], Nitin Kapadia[1†], Emilien Nicolas[2‡], Stephan Uphoff[2], Adam JM Wollman[3], Mark C Leake[3], Rodrigo Reyes-Lamothe[1*]**

[1]Department of Biology, McGill University, Montreal, Canada; [2]Department of Biochemistry, University of Oxford, Oxford, United Kingdom; [3]Biological Physical Sciences Institute, Departments of Physics and Biology, University of York, Heslington, United Kingdom

**Abstract** The replisome is a multiprotein machine that carries out DNA replication. In *Escherichia coli*, a single pair of replisomes is responsible for duplicating the entire 4.6 Mbp circular chromosome. In vitro studies of reconstituted *E. coli* replisomes have attributed this remarkable processivity to the high stability of the replisome once assembled on DNA. By examining replisomes in live *E. coli* with fluorescence microscopy, we found that the Pol III* subassembly frequently disengages from the replisome during DNA synthesis and exchanges with free copies from solution. In contrast, the DnaB helicase associates stably with the replication fork, providing the molecular basis for how the *E. coli* replisome can maintain high processivity and yet possess the flexibility to bypass obstructions in template DNA. Our data challenges the widely-accepted semi-discontinuous model of chromosomal replication, instead supporting a fully discontinuous mechanism in which synthesis of both leading and lagging strands is frequently interrupted.

*For correspondence: rodrigo.
reyes@mcgill.ca

†These authors contributed
equally to this work

Present address: ‡Fast Track
Diagnostics Luxembourg S.à.r.l,
Luxembourg City, Luxembourg

**Competing interests:** The
authors declare that no
competing interests exist.

**Reviewing editor:** James M
Berger, Johns Hopkins University
School of Medicine, United
States

## Introduction

DNA replication is carried out by a multifunctional machine, the replisome (***Beattie and Reyes-Lamothe, 2015***). The *E. coli* replisome has been characterized in vitro and in vivo and is composed of more than 12 different proteins (***Kurth and O'Donnell, 2013***; ***Reyes-Lamothe et al., 2010***). DNA synthesis is performed by the Pol III polymerase ($\alpha\epsilon\theta$). Three copies of Pol III are incorporated into the replisome through an interaction with the $\tau$ subunit of the pentameric clamp loader complex ($\tau_3\delta\delta'$). Together, these constitute the Pol III* subassembly (($\alpha\epsilon\theta)_3$-$\tau_3\delta\delta'$). The clamp loader is also responsible for loading the $\beta$ clamp dimer onto DNA, which is required for processive synthesis by Pol III. Addition of $\beta$ clamp to Pol III* forms the Pol III holoenzyme. At the core of the *E. coli* replisome is the replicative helicase, DnaB, which encircles the lagging strand template and unwinds parental DNA. The Pol III holoenzyme associates with DnaB through the $\tau$ subunit of the clamp loader (***Figure 1A***). In addition, the DnaB helicase recruits the primase, DnaG, which synthesizes RNA primers. Due to the antiparallel nature of DNA, synthesis of one the strands – the leading strand – occurs co-directionally with progression of the replication fork, while the second strand – the lagging strand – is synthesized by repeated cycles of primer synthesis and DNA extension.

Replication of the circular chromosome of *E. coli* proceeds bidirectionally from a single, defined locus: *oriC*. Multiple mechanisms tightly restrict DnaB loading, and therefore replisome assembly, to the *oriC* locus during initiation, with a single initiation event per cell cycle (***Costa et al., 2013***). Two sister replisomes are assembled at *oriC* during initiation, and each is responsible for replicating half

**eLife digest** New cells are created when an existing cell divides to produce two new ones. During this process the original cell must copy its DNA so each new cell inherits a full set of genetic material. DNA is made up of two strands that twist together to form a double helix. These strands need to be separated so they can be used as templates to make new DNA strands. An enzyme called DNA helicase is responsible for separating the two DNA strands and another enzyme makes the new DNA. These enzymes are part of a group of proteins collectively called the replisome that controls the whole DNA copying process.

The replisome must be extremely reliable to avoid introducing mistakes into the cell's genes. Previous research using replisomes extracted from cells indicated that replisomes are effective at copying DNA because the proteins they contain are strongly bound together and remain attached to the DNA for a long time. However, the behavior of replisomes in living cells has not been closely examined.

Beattie, Kapadia et al. used microscopy to observe how the replisome copies DNA in a bacterium called *Escherichia coli*. The experiments revealed that most of the proteins within the replisome are constantly being replaced during DNA copying. The exception to this is DNA helicase, which stays in place at the front of the replisome, providing a landing platform for all the other parts of the machine to come and go.

Future work will investigate why the parts of the replisome are replaced so frequently. This may allow us to alter the stability of the bacterial replisome, which may lead to new medical treatments and biotechnologies.

of the ~4.6 Mbp chromosome. At 37°C, it takes 40–60 min of continuous DNA synthesis to complete chromosomal replication, at rates of 600–1000 bp s$^{-1}$.

Given the extent of DNA synthesis required, it has been assumed that the replisome is a stable protein complex capable of replicating large fragments of the chromosome without disassembling. This is supported by in vitro data showing that a single purified replisome, once assembled on DNA, is capable of synthesizing DNA with an average length of 70 kbp without requiring replacement of the Pol III* subassembly or DnaB (*Tanner et al., 2011*; *Yao et al., 2009*). Even greater stability has been inferred from in vivo experiments that suggest infrequent replication fork collapse during chromosome replication in *E. coli* (*Maisnier-Patin et al., 2001*).

Chromosomal DNA presents multiple potential obstacles to replisome progression. DNA lesions can result in the stalling of the replisome due to Pol III's inability to use damaged DNA as a template (*Moore et al., 1981*). In addition, the replisome frequently encounters DNA-bound proteins, potentially resulting in disassembly or pausing (*Mettrick and Grainge, 2016*; *Gupta et al., 2013*). Multiple mechanisms have been proposed that allow replisome integrity to be maintained during bypass of such obstacles (*Heller and Marians, 2006a*; *Yeeles and Marians, 2011*; *Pomerantz and O'Donnell, 2008*) and that remove bound proteins from DNA (*Gupta et al., 2013*). In cases where these strategies are insufficient, the cell also has mechanisms to mediate the reassembly of the replisome at specific DNA structures that arise following replisome collapse (*Heller and Marians, 2006b*). The frequency at which replisomes encounter these obstacles and the efficiency of the bypass mechanisms are still unclear. It is also uncertain in which way the architecture and stability of replisome play a role during these events.

The replisome is likely to be affected by multiple factors present inside cells which have not been accounted for in reconstituted systems. However, a direct measurement of the stability of the replisome has not been undertaken in vivo. Here we measure the binding kinetics of replisome subunits during DNA replication using two independent fluorescence-based methods in living cells. Our results show that the entire Pol III* subassembly is replaced within the replisome at a frequency equivalent to a few cycles of Okazaki fragment synthesis. This leads us to conclude that DNA replication is a discontinuous process on both strands. We also find that the DnaB helicase remains bound to DNA for tens of minutes, preventing disassembly of the replisome likely by serving as a dock during Pol III* subassembly turnover. We propose that this dynamic stability provides the replisome

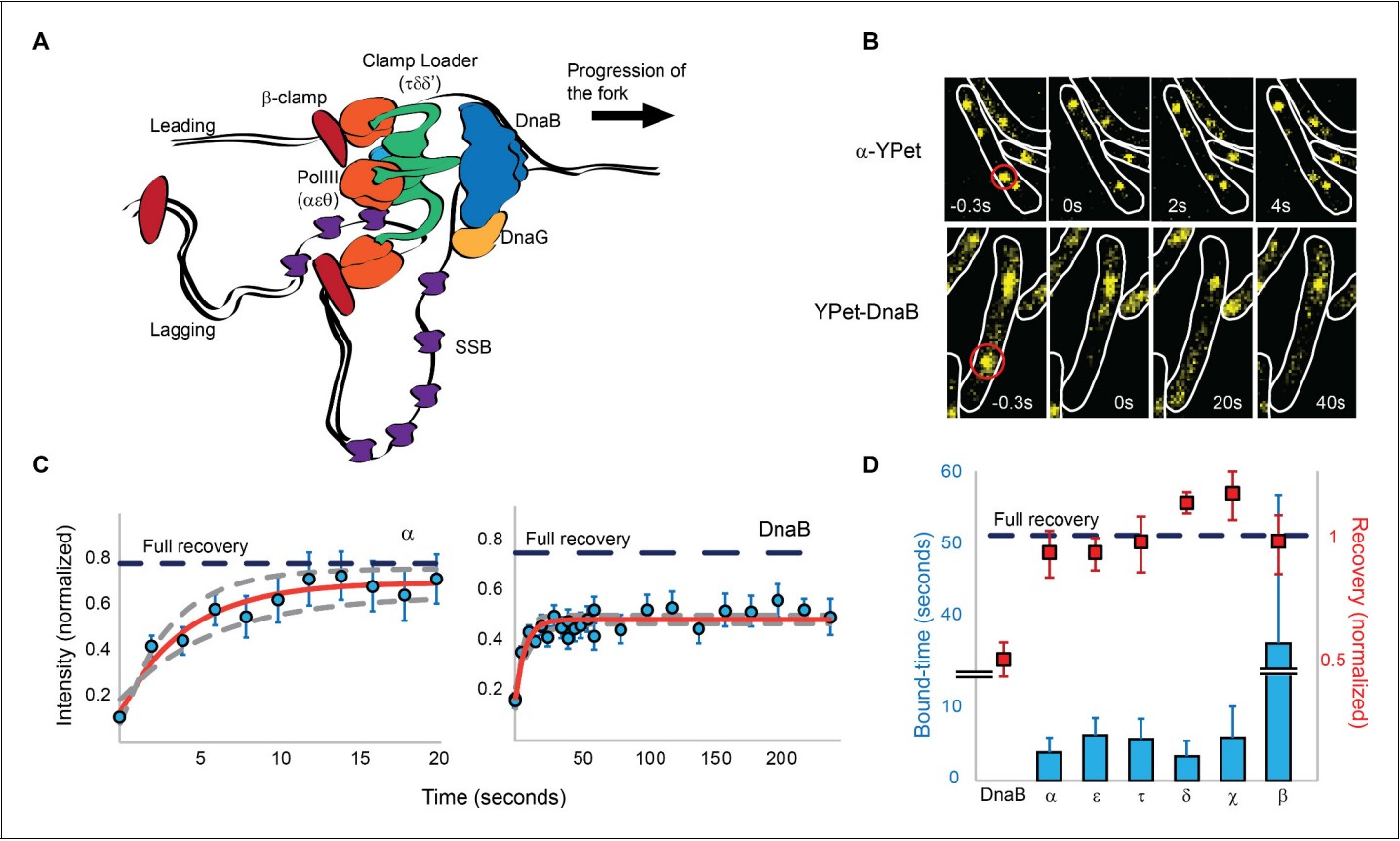

**Figure 1.** Most replisome subunits exchange frequently with the diffusing pool. (**A**) Model illustrating the architecture of a replisome at the *E. coli* replication fork. (**B**) Representative fluorescence images of FRAP experiments for the Pol III α subunit and the DnaB helicase. Cell boundaries shown as white lines, red circle shows the location of the bleached focus. (**C**) Representative examples of the FRAP curves for Pol III α subunit (N = 48) and DnaB (N = 96). Red line shows a reaction-diffusion model fit to the data, dashed grey lines show SE for the model. Dashed blue line represents the estimated maximum possible fluorescence recovery after correcting for photobleaching. (**D**) Analysis summary of the replisome by FRAP. Bars represent average bound-times. Red squares represent level of recovery normalised to the intensity before bleaching. Dashed blue line represents maximum possible fluorescence recovery. It was not possible to estimate the bound-time for DnaB. Error bars represent SE.

The following figure supplements are available for figure 1:

**Figure supplement 1.** Artificial elongation of cells by cephalexin treatment does not interfere with DNA replication or protein expression.

**Figure supplement 2.** Minimal contribution of YPet photoblinking during FRAP.

**Figure supplement 3.** Growth rate and replication time of *E. coli* in our experimental conditions.

with flexibility to bypass frequent obstacles on DNA while maintaining the necessary processivity for chromosomal replication.

## Results

### Fluorescence recovery after photobleaching reveals frequent exchange of subunits in active replisomes

To assess the stability of the *E. coli* replisome when replicating chromosomal DNA in vivo, we first measured the binding kinetics of replisome subunits using fluorescence recovery after photobleaching (FRAP) in strains possessing fluorescent YPet derivatives of key replisome components (*Reyes-Lamothe et al., 2010*, *2008*). Using actively replicating cells in growth conditions that permit a

single replication event per cell cycle, we bleached individual foci of fluorescent replisome subunits using a focused laser pulse and measured their recovery over time (*Figure 1B*). The dimensions of *E. coli* – in our conditions typically ~0.7 µm diameter and few microns in length – and the low number of replisome subunit molecules per cell – a few hundred for most subunits (*Reyes-Lamothe et al., 2010*) – increased the difficulty of selectively bleaching replisome foci without affecting the remaining fluorescent pool. To minimize photobleaching of the diffusing pool of fluorescent proteins we increased cell volume by treatment with cephalexin; this did not affect DNA replication (*Figure 1—figure supplement 1*).

To our surprise, we found that the initial focus fluorescence recovered in a few seconds for Pol III and clamp loader components (*Figure 1B*). Fluorescence recovery is not explained by the photophysical properties of YPet, like photoblinking, and instead it represents protein exchange (*Figure 1—figure supplement 2*). We used a reaction-diffusion model in a reaction-limited regime to fit the average fluorescence recovery curve of individual subunits, and calculated a time constant for binding (bound-time) which represents the average time that a molecule is bound to the replisome before exchanging. The bound-time was $4 \pm 2$ and $6 \pm 2$ s (mean ± SE) for the $\alpha$ and $\varepsilon$ subunits of Pol III, respectively. Similarly, the $\tau$, $\delta$ and $\chi$ subunits of the clamp loader had bound times of $6 \pm 3$, $3 \pm 2$ and $6 \pm 4$ s, respectively (*Figure 1C–D*). Molecules of the $\beta$ clamp exchanged at a slower rate, remaining associated for an average of $36 \pm 21$ s, consistent with its binding to newly-synthesized DNA behind the replisome (*Moolman et al., 2014*; *Su'etsugu and Errington, 2011*). The timescale of Pol III holoenzyme exchange is in striking contrast to the ~150 min required for two replisomes to complete chromosomal replication under our microscopy conditions (*Figure 1—figure supplement 3*).

## sptPALM demonstrates fast turnover of the Pol III* subassembly

To confirm these results, we used single-particle tracking Photoactivated Localization Microscopy (sptPALM) (*Manley et al., 2008*) to determine the bound-times of replisome subunits (*Uphoff et al., 2013*). We constructed *E. coli* strains with functional fusions of replisome subunits and the photoconvertible fluorescent protein, mMaple (*McEvoy et al., 2012*) (*Figure 2—figure supplement 1*). We used a single low intensity pulse of 405 nm-laser activation per experiment to switch, on average, a single molecule per cell into a red fluorescence state. Long (500 ms) camera exposure times – to motion-blur fast diffusing molecules – were used, spaced by 1 s or 5 s intervals, to track non-diffusing replisome-associated molecules as foci (*Figure 2A*). This illumination protocol did not perturb cell growth (*Figure 2—figure supplement 2*). Track duration distributions for labelled replisome subunits were calculated from the number of frames individual molecules appeared as foci (*Figure 2B–C*). Bound times were calculated by correcting for the disappearance of foci due to photobleaching, which was characterized using a strain carrying the transcriptional repressor LacI fused to mMaple and a chromosomal array of *lacO* binding sites. We also assessed the effect photoblinking using this same strain (*Figure 2—figure supplement 3A–C*). We expect that in the timescale of our experiments, the lifetime of LacI-mMaple foci will be dictated by photobleaching, with dissociation from DNA being negligible (*Hammar et al., 2014*).

The single-molecule results are consistent with our FRAP data. Pol III subunit and clamp loader components indeed exchanged rapidly, with $\varepsilon$, $\tau$ and $\delta$ remaining replisome-associated for only $10 \pm 0.7$, $10 \pm 0.7$ and $12 \pm 0.9$ s (mean ± SE), respectively (*Figure 2B and D*). We found no strong evidence for multiple binding behaviors of individual subunits (*Figure 2—figure supplement 3D* and *Supplementary file 1C*), suggesting that both leading and lagging strand polymerases behave similarly. As with FRAP, we observed similar bound-times for all subunits of both the DNA polymerase III and clamp loader complexes despite a difference in stoichiometry – $\delta$, $\tau$ and $\varepsilon$ are present in 1, 3 and 3 copies per replisome, respectively (*Figure 1A*). As such, exchange of individual subunits independently from one another, although still possible, does not easily explain our results. We therefore propose that the unit of exchange of Pol III and clamp loader subunits is the Pol III* subassembly $((\alpha\varepsilon\theta)_3\text{-}\tau_3\delta\delta')$. This idea is supported by in vitro data that shows that exchange of Pol III at the replication fork requires it to be part of the Pol III* subassembly (*Yuan et al., 2016*). Cells have an excess of free Pol III that does not interact with the clamp loader (*Maki and Kornberg, 1985*). Consistent with the notion that only Pol III subunits found within a Pol III* subassembly are competent for exchange, we observed re-binding of single molecules of $\varepsilon$, often to a different replisome, at a much higher frequency than would be predicted if all of the ~270 molecules of $\varepsilon$ in the cell were

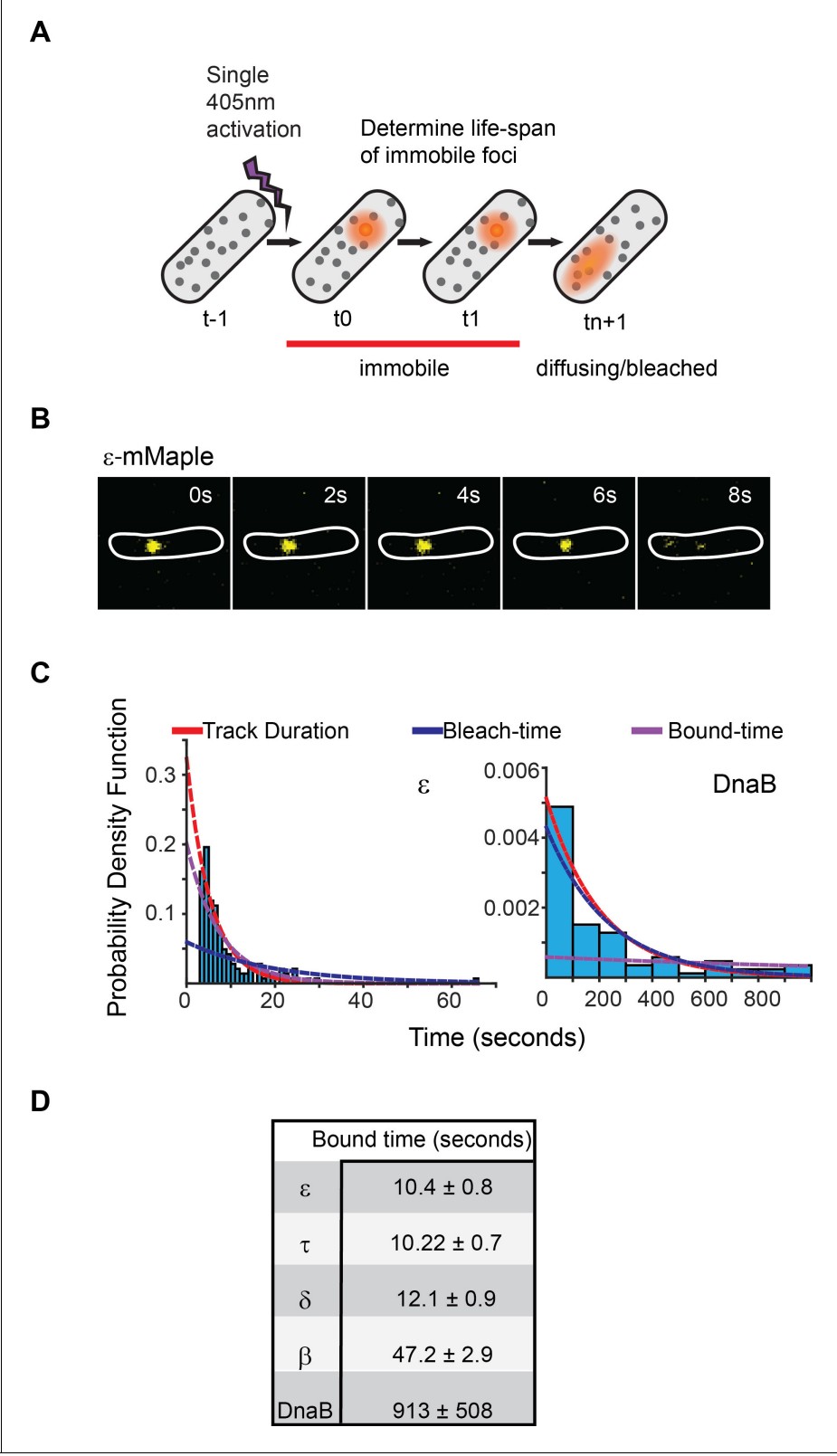

**Figure 2.** Exchange of Pol III* subassembly and DnaB occur on different timescales. (**A**) Diagram illustrating the sptPALM experimental design used to measure bound-times. (**B**) Representative example of the focus life span for the Pol III ε subunit. (**C**) Representative examples of the distribution of fluorescent foci life-spans (blue bars) for Pol III ε subunit and DnaB, showing fitting of a single-exponential decay model (red line), the estimated bleaching rate in the same conditions (blue line) and the corrected estimated bound-time (purple line). Note that to improve accuracy in single-

*Figure 2 continued on next page*

*Figure 2 continued*

molecule detection tracks shorter than four localizations were removed in the case of ε but corrected during curve fitting, hence the lower bar near 0 s time point. ε data was collected using 500 ms exposure time and 1 s intervals (N = 143), DnaB data was collected using 2 s exposure time and 10 s intervals (N = 86). The plot for DnaB shows binned data for presentation purposes. (D) Summary of estimated average bound-times. Errors in the table represent SE.

The following figure supplements are available for figure 2:

**Figure supplement 1.** Characterisation of mMaple fusions.

**Figure supplement 2.** Minimal exposure to 405 nm activation light allows continuation of cell growth.

**Figure supplement 3.** Estimation of photoblinking, test for two binding kinetic regimes and characterisation of the effect of longer 2 s capture rates in our estimation of bound-times.

**Figure supplement 4.** Slow diffusion of DnaB helicase complicates correct assignment of immobile molecules at sub-second capture rates.

in direct competition (*Reyes-Lamothe et al., 2010*) (*Figure 3A* and *Figure 3—figure supplement 1*).

The β clamp again showed a longer bound-time of 47 ± 3 s in sptPALM. Our estimate is broadly consistent with a previous estimate from *E. coli*, although ~4 times shorter (*Moolman et al., 2014*). Assuming similar numbers of DNA-bound β as in that earlier report, we estimate from our results that a new β dimer is loaded at each replication fork every ~2 s. Considering an average rate of fork progression of ~260 bp s$^{-1}$ – calculated from the duration of replication in the conditions used (*Figure 1—figure supplement 3*) – we find an average Okazaki fragment length of 520 bp, in close agreement with an in vitro measurement of 650 bp at room temperature (*Yao et al., 2009*). In contrast, we estimate from FRAP and sptPALM an average replication fork progression of 1–3 kbp prior to Pol III exchange, which would allow completion of multiple Okazaki cycles. We therefore think it unlikely that subunit exchange within the replisome is exclusively linked to the dynamics on the lagging strand.

## DnaB may act as a stable platform upon which the Pol III* subassembly exchanges

The DnaB helicase displayed very different dynamics to other replisome subunits when assessed by FRAP. Crucially, we never observed full fluorescence recovery of DnaB over 5 min of measurement (*Figure 1B–D*), indicating that it is stably associated with the replisome on this timescale. Our analysis showed an initial recovery of fluorescence with a 7 ± 4 s time constant, which we attribute to the signal from diffusing molecules moving into the bleached area. However, we did not observe a significant increase in the intensity after this initial time point, preventing us from accurately estimating a bound-time for DnaB (*Figure 1C*). We conclude that in contrast to Pol III holoenzyme subunits, replisome-associated DnaB does not exchange frequently, and is instead a stable component of the replisome.

Analysis of DnaB by sptPALM confirmed that it is the most stable subunit in the replisome. To eliminate incorrect assignment of DnaB fluorescence by our software we used even longer (2 s) exposure times, thus further blurring slow-diffusing molecules (*Figure 2—figure supplement 4*). Control experiments with ε under the same conditions had no significant effect on calculated bound-times (*Figure 2—figure supplement 3E*). Crucially, we estimate that single molecules of DnaB remained bound to the replisome for 913 ± 508 s, significantly longer than any other component (*Figure 2B–D*). The width on the distribution for this estimate is inherently large due to a close similarity between DnaB foci lifetimes and the bleaching time of mMaple (*Figure 2C*). However, our long bound-time estimate is supported by frequent examples of DnaB fluorescent foci that last for tens of minutes (*Figure 3B*). Currently we cannot assess the extent at which the turnover detected represents PriABC mediated re-loading of helicase (*Heller and Marians, 2006b*). Altogether, our data supports a role for DnaB as the primary determinant of replisome integrity.

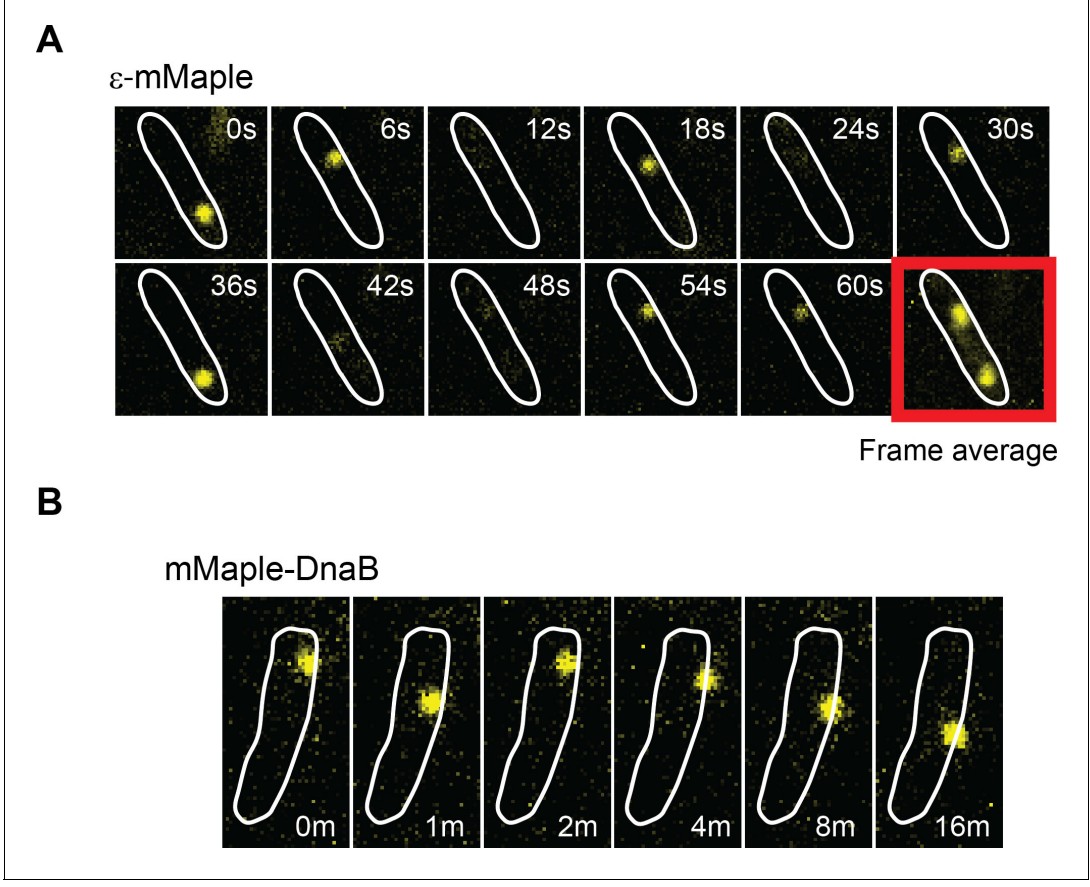

**Figure 3.** DnaB may serve as a platform for frequent Pol III* subassembly re-binding. (**A**) Representative example of a cell where a single activate copy of ε-mMaple shows multiple cycles of binding and unbinding (time in seconds). Obtained from an experiment using 2 s intervals between consecutive images. Frame average shows the cellular location of two replisomes. (**B**) Example of a cell where a single activated mMaple-DnaB molecule remains localized as a focus for several minutes (time in minutes). Obtained from an experiment using 10 s intervals between consecutive images. Boundaries of the cells at the beginning of the experiment are shown as white outlines.

The following figure supplement is available for figure 3:

**Figure supplement 1.** Re-binding of copies of ε at the same position are unexpectedly frequent.

## Active synthesis is only partially responsible for turnover

To determine if subunit exchange occurs as a consequence of active DNA synthesis, we measured bound-times by FRAP and sptPALM in cells treated with the DNA polymerase inhibitor hydroxyurea (HU) (*Sinha and Snustad, 1972*). Components of the Pol III* subassembly and the β-clamp showed bound-times two- to six fold higher than in untreated cells (*Figure 4A–B*). HU had no apparent effect on DnaB FRAP estimates (*Figure 4A*). We conclude that the exchange of replisome components is at least partially dependent on active DNA synthesis. Remaining turnover may reflect residual DNA synthesis after HU treatment. In addition, it may indicate that the replisome is intrinsically dynamic as a multiprotein complex, with DNA synthesis further increasing subunit exchange.

## Discussion

### A dynamic replisome may help to minimize delays in replication fork progression in the presence of roadblocks

Obstructions in template DNA, particularly protein-DNA complexes, have been shown to cause frequent pausing of the *E. coli* replisome in vivo (*Gupta et al., 2013*). In vitro studies have

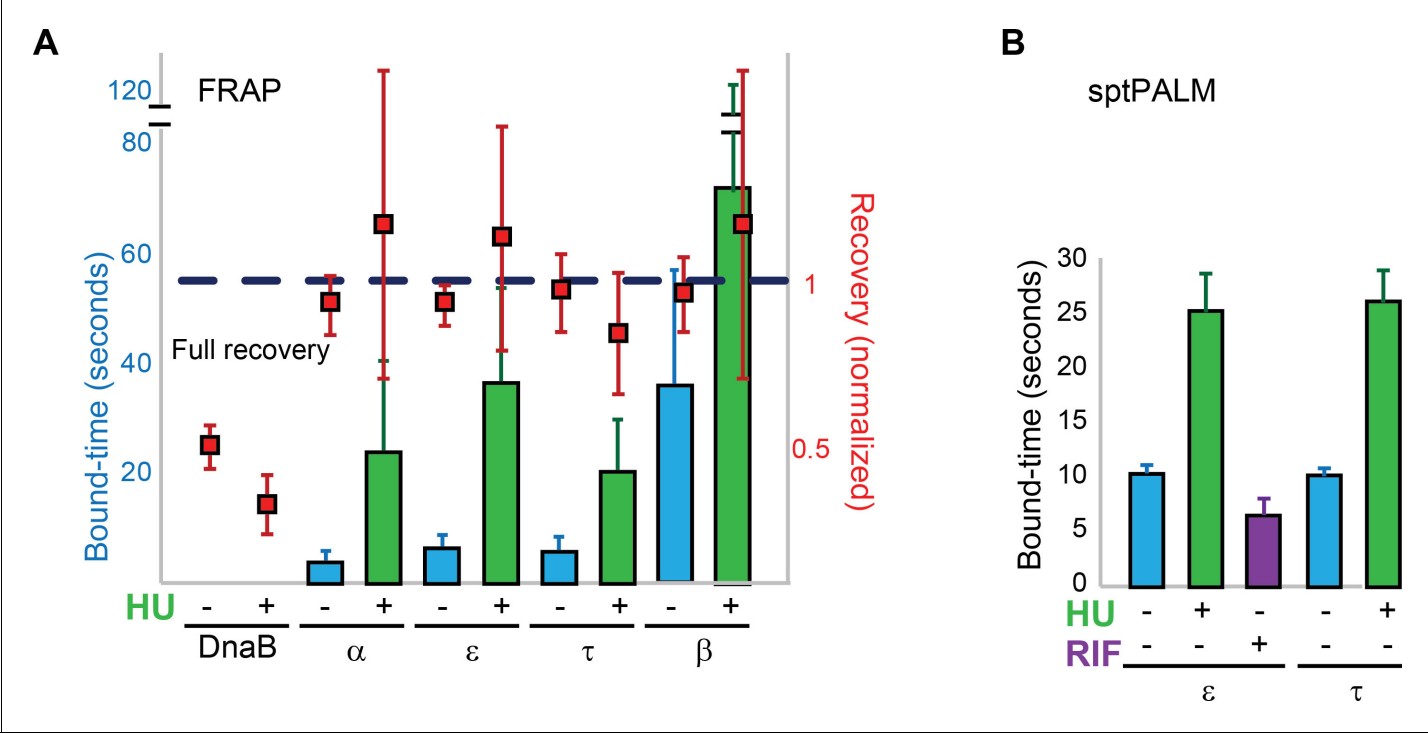

**Figure 4.** Replisome dynamics are partly dependent on active DNA synthesis. (**A**) Summary of the average bound-times in cells treated with HU, estimated by FRAP. The results for untreated (blue bars) and treated cells (green bars) is shown. Data for the untreated condition is presented to facilitate comparison and is identical to that in *Figure 1D*. Red squares represent the normalised level of fluorescence recovery. Dashed blue line shows estimated maximum possible recovery. It was not possible to estimate the bound-time for DnaB. (**B**) Summary of bound-times estimated by sptPALM (weighted average). Results for the untreated (blue bars) and treated cells (green bars) is shown. Data obtained using cells treated with the RNA Polymerase inhibitor Rifampicin is also shown for cells carrying ε-mMaple (purple bars). Data for the untreated condition is presented to facilitate comparison and is identical to that in *Figure 2D*. Error bars represent SE.

demonstrated that the *E. coli* replisome is capable of bypassing such obstacles by interrupting leading strand synthesis and resuming extension downstream of the obstruction from a leading strand primer deposited by DnaG or an mRNA synthesized by RNA polymerase (*Heller and Marians, 2006a*; *Yeeles and Marians, 2011*; *Pomerantz and O'Donnell, 2008*). Our data is entirely consistent with this mechanism, whereby bypass could be achieved through detachment of the stalled Pol III* subassembly from DnaB and its replacement downstream of the obstacle with another Pol III* from solution. Because DnaB translocates on the lagging strand, small lesions and large protein blockages on the leading strand can both be bypassed by Pol III* exchange. Note however that we did not observe any apparent effect on the dynamics of Pol III after inhibiting transcription, suggesting that this process is not the main cause of exchange (*Figure 4B*). On the lagging strand, small template lesions capable of passing through the central pore of DnaB may also be bypassed through Pol III* exchange. In contrast, obstacles on the lagging strand that destabilize DnaB, such as proteins stably bound to DNA or strand discontinuities, would likely result in the disassembly of the replisome. We propose that obstacle bypass along template DNA may be the primary selection pressure that has driven the evolution of a dynamic replisome.

In addition to the model above, we acknowledge that other processes may also exert selective pressure for the generation of the observed replisome binding kinetics. First, unbinding of Pol III* subassembly may result from build-up of helical torsion in the template DNA generated by the coupled synthesis of both DNA strands (*Kurth et al., 2013*). This is consistent with longer binding times when synthesis was inhibited by HU. Unbinding of a single polymerase from DNA or release of the whole Pol III* subassembly would have the same effect on stress relief. Second, the dynamics observed may be a byproduct of the highly regulated interaction between Pol III and β-clamp. Even

**Table 1.** Strains used for this study.

| Strain | Relevant genotype | Source |
|---|---|---|
| AB1157 | thr-1, araC14, leuB6(Am), DE(gpt-proA)62, lacY1, tsx-33, qsr'-0, glnV44(AS), galK2(Oc), LAM-, Rac-0, hisG4(Oc), rfbC1, mgl-51, rpoS396(Am), rpsL31(strR), kdgK51, xylA5, mtl-1, argE3(Oc), thi-1 | *Dewitt and Adelberg, 1962* |
| RRL27 | holC-ypet kan | *Reyes-Lamothe et al. (2008)* |
| RRL30 | holE-ypet kan | *Reyes-Lamothe et al. (2008)* |
| RRL32 | ssb-ypet kan | *Reyes-Lamothe et al. (2008)* |
| RRL33 | holA-ypet kan | *Reyes-Lamothe et al. (2008)* |
| RRL34 | holD-ypet kan | *Reyes-Lamothe et al. (2008)* |
| RRL35 | dnaE-ypet kan | *Reyes-Lamothe et al. (2008)* |
| RRL36 | dnaQ-ypet kan | *Reyes-Lamothe et al. (2008)* |
| RRL51 | dnaX-ypet kan | *Reyes-Lamothe et al. (2008)* |
| RRL196 | frt ypet-dnaN | *Reyes-Lamothe et al. (2010)* |
| RRL368 | frt-ypet-dnaB | *Reyes-Lamothe et al. (2010)* |
| RRL537 | dnaQ-mMaple kan | This study |
| RRL538 | holA-mMaple kan | This study |
| RRL541 | tetR-ypet kan, [tetO240-gm]852 | This study |
| RRL553 | dnaX-mMaple kan | This study |
| RRL557 | frt mMaple-dnaB | This study |
| RRL558 | frt mMaple-dnaN | This study |
| TB44 | dnaB-mMaple kan | This study |
| TB54 | lacI-mMaple kan, [lacO240-hyg]2735::ΔpheA | This study |
| Plasmid | Features | Source |
| pKD46 | Expression of lambda red genes | *Datsenko and Wanner, 2000* |
| pCP20 | Expression of Flp recombinase | *Datsenko and Wanner, 2000* |
| pROD61 | mYPet Kan R6K gamma ori. For C-ter insertions | This study |
| pROD83 | YPet Kan R6K gamma ori. For N-ter insertions | This study |
| pROD93 | mMaple Kan R6K gamma ori. For C-ter insertions | This study |
| pROD160 | mMaple Kan R6K gamma ori. For N-ter insertions | This study |

though Pol III tightly binds the $\beta$-clamp to ensure highly processive synthesis, these two proteins rapidly unbind from each other upon completion of the duplex DNA. The strength of this protein-protein interaction is modulated by the OB domain in the $\alpha$ subunit of Pol III, which binds to ssDNA ahead of the catalytic domain, and the C-terminus of the $\tau$ subunit of the clamp loader (*Georgescu et al., 2009*, *Leu et al., 2003*). Premature activation of such a switch in both leading and lagging strand polymerases would weaken the grip of the Pol III* subassembly on the replication fork and potentially result in its displacement. This idea is consistent with the presence of ssDNA gaps in the lagging strand (*Li and Marians, 2000*), which may be explained by a premature loss of Pol III processivity.

Presumably, exchange of Pol III* subassembly within the replisome occurs rapidly enough to minimize potentially deleterious ssDNA gaps between fragments of nascent DNA on the leading strand. However, the rate of DNA unwinding by DnaB decreases by more than 10-fold when DnaB is detached from the $\tau$ subunit of the clamp loader (*Kim et al., 1996a*), providing a potential safety mechanism to limit DNA unwinding and exposure of ssDNA during Pol III* subassembly exchange. It

remains to be determined if a newly associated copy of Pol III uses the existing 3' end at the leading strand or requires the activity of primase to resume synthesis.

## Protein excess in cells is a key factor in the regulation of replisome subunit turnover

Our data apparently contradict in vitro studies which have demonstrated that a single reconstituted *E. coli* replisome can operate without subunit exchange in synthesizing an average of ~80 kbp (*Tanner et al., 2011*; *Yao et al., 2009*). Measurements in those reports were performed by removing all diffusing Pol III* subassembly and DnaB subunits from the reaction. In contrast, in the cell there is a permanent excess of diffusing replisome subunits. We believe this explains the differences observed with our in vivo data. Competition for binding sites between DNA-bound and diffusing molecules has been shown to change the DNA-binding kinetics of proteins such as Fis, HU and NHP6A (*Graham et al., 2011*), EcoRI (*Sidorova et al., 2013*), RPA (*Gibb et al., 2014*) and the transcription factor CueR (*Chen et al., 2015*). Furthermore, mathematical modelling has shown that it is theoretically possible for a replisome to be stable under conditions in which no extra subunits are present, as in vitro, and yet undergo frequent subunit exchange in the presence of extra subunits, as in vivo, due to subunit competition (*Åberg et al., 2016*). We think that active synthesis may enhance exchange with the diffusing pool, consistent with our results using HU.

Frequent exchange of DNA polymerases in the presence of extra subunits has been observed in the replisomes of bacteriophages T4 and T7 in vitro (*Yang et al., 2004*; *Johnson et al., 2007*). In T7, this occurs through a mechanism in which extra DNA polymerases associate with the bacteriophage DNA helicase and exchange with the active DNA polymerase through competition for DNA binding (*Geertsema et al., 2014*; *Loparo et al., 2011*). It will be interesting in the future to determine the mechanisms that exist in *E. coli* to ensure efficient capture and exchange of the low-abundance Pol III* subassembly.

## DNA synthesis of both leading and lagging strands is discontinuous in *E. coli*

One predicted consequence of the Pol III* subassembly exchanging as a single unit is that synthesis of both leading and lagging strands will be frequently interrupted, resulting in discontinuities on both strands. This contrasts the widely-accepted semi-discontinuous model of DNA replication. However, while this model is strongly supported by in vitro experiments (*Wu et al., 1992*), the mechanism that operates in vivo has long been unclear (*Yeeles, 2014*; *Wang, 2005*). Okazaki and colleagues' original characterization of replication intermediates demonstrated that all DNA is initially synthesized as short fragments, supporting fully discontinuous DNA replication (*Okazaki et al., 1968*). More recent in vivo experiments performed in the absence of DNA ligase support the idea that discontinuities are produced on both leading and lagging strands during DNA replication in *E. coli* (*Amado and Kuzminov, 2013*, *2006*). Our data provide a mechanistic explanation for these observations, and supports a discontinuous model of DNA synthesis in *E. coli*.

## Materials and methods

### Strains and growth conditions

All strains used are derivatives of AB1157. Cells were routinely grown in LB or in M9 minimal media. M9 was supplemented with glycerol (final concentration 0.2%); 100 μg/ml of amino acids threonine, leucine, proline, histidine and arginine; and thiamine (0.5 μg/ml). When required, antibiotics were added at the following concentrations: ampicillin (100 μg/ml), kanamycin (30 μg/ml), chloramphenicol (25 μg/ml), cephalexin (40 μg/ml), rifampicin (Rif) (300 μg/ml) and hydroxyurea (HU) (60–100 mM). For microscopy, cells were spotted on a 1% agarose pad in M9-Glycerol. DAPI was used at a working concentration of 300 nM as recommended by manufacturer. Ethanol fixation was done using 70% ethanol in water, followed by two washes with PBS. For TetR-YPet strain, fixation was done using 4% formaldehyde and incubating 15 min at room temperature, 15 min on ice, followed by two washes with PBS.

Chromosomal replacement of replisome genes by fluorescent derivatives was done by lambda red (*Reyes-Lamothe et al., 2008*; *Datsenko and Wanner, 2000*). In short, we used plasmids

carrying a copy of *ypet* (*Reyes-Lamothe et al., 2008*; *Nguyen and Daugherty, 2005*) or *mMaple* (*McEvoy et al., 2012*) followed or preceded by a kanamycin resistance cassette flanked by *frt* sites as PCR templates. Flexible peptides with sequences SAGSAAGSGEF (YPet C-ter fusions), SAG-SAAGSGAV (mMaple C-ter fusions) or SAGSAAGSGSA (YPet and mMaple N-ter fusions) were used as a linker between the fluorescent protein (FP) and the protein targeted. Primers carrying 40-50nt tails with identical sequence to the chromosomal locus for insertion were used to amplify the *linker-FP-kan^R* (or *kan^R-FP-linker* in the case of N-terminal fusions) from template plasmids. The resulting PCR product was transformed by electroporation into a strain carrying the lambda red-expressing plasmid pKD46. Colonies were selected by kanamycin resistance and ampicillin sensitivity, screened by PCR using primers annealing to regions flanking the insertion, and sequenced. In the case of N-terminal fusions, in order to minimize the effect of the insertion on the expression levels of the gene we removed the kanamycin cassette by expressing the Flp recombinase from plasmid pCP20 (*Datsenko and Wanner, 2000*). Gene fusions did not have any apparent detrimental effect on cell growth (*Figure 2—figure supplement 1*).

LacI-mMaple was generated through lambda red using the strain MG1655. The gene fusion was then transduced, using P1 phage, into an AB1157 derivative carrying a 256-*lacO* array replacing the *pheA* gene (chromosomal position 2735 kb)(*Wang et al., 2006*). Similarly, a TetR-YPet fusion expressed from a lac promoter (*Reyes-Lamothe et al., 2014*) was transduced into a strain carrying a 256-*tetO* operator array at R3 (chromosomal position 852 kb) (*Wang et al., 2006*).

## Fluorescence recovery after photobleaching (FRAP)

Cells were grown in M9-Glycerol at 30°C, treated with cephalexin for 2 hr, harvested at early log-phase ($OD_{600}$0.1–0.2), concentrated and spotted onto a pad of 1% agarose in M9-Glycerol, contained in a gene frame (Thermo Scientific). Treatment with hydroxyurea was done on the agarose pad by mixing HU with media and agarose. Cells were incubated on the slides for 10 min before imaging.

Most FRAP experiments, except for the TetR-YPet control, were performed using a spinning disk imaging system (PerkinElmer) with a 100x NA1.35 oil objective and an ImagEM EMCCD camera (Hamamatsu Photonics). Images were acquired using Volocity imaging software. An image was acquired in the brightfield channel at the beginning of the experiment to serve as a reference. FRAP was performed by pulse-bleaching using a 488 nm laser for 10–15 ms and 30–50% laser intensity (radius of the spot was diffraction limited at ~300 nm). Two pre-bleach images were captured, the bleach spot was centered on one replisome focus and recovery of the bleached region was recorded at different intervals after bleaching. Image capture was done at a 300 ms frame rate (4–6% 515 nm laser) for most replisome components except for DnaB helicase, for which 500 ms capture rate was used (2% 515 nm laser). For $\alpha$, $\varepsilon$, $\tau$, $\delta$ and $\chi$, we used intervals between pictures of 2 s, 5 s and 10 s. For DnaB and $\beta$, we used intervals between pictures of 5 s, 10 s and 20 s. Experiments were done at room temperature.

FRAP to control for photoblinking was done using an epifluorescence system, Leica DMi8, with a 100x oil objective (Leica 100x/NA 1.47 HL PL APO) and an iXon Ultra 897 EMCCD camera (Andor). FRAP was performed using an iLas2 unit (Roper Scientific) using an ILE laser combiner (Andor) and a 150 mW 488 nm laser. Both bleaching and excitation of YPet were done using the 488 nm laser. Acquisition was done using 100 ms exposure at 5 s intervals.

## FRAP analysis

Initial position of spots was manually selected using the coordinates for localized bleaching in the image recorded by the acquisition software. Tracking was then done automatically using a previously developed custom program in MATLAB (Mathworks), ADEMS code (*Miller et al., 2015*) (freely available at https://sourceforge.net/projects/york-biophysics/). Most experiments analyzed had a pixel size of 100 nm, for which we used a search window with a radius of 5 pixels and an initial guess for the PSF of 3 pixels when fitting candidate spots. For a minority of the experiments, with pixel size of 140 nm, analysis was done using 4-pixel search window and a 2-pixel radius for initial fitting.

Intensity traces were filtered to retain only those where clear bleaching was observed. We removed any trace where the intensity at any of the pre-bleach time points was below the value of the ROI immediately after bleaching (0 s time point). In addition, the intensity at the 0 s time point

had to be below 40% of the mean pre-bleach intensity. FRAP data were then normalized by the average intensity of the pre-bleached data points.

To estimate the maximum possible fluorescence recovery (Max recovery), we used the corresponding brightfield image to draw a polygon in ImageJ (*Schneider et al., 2012*) around cells containing a bleached spot. We used these ROIs to obtain the intensities across the experiment in the fluorescence channel. Max recovery was calculated by dividing the intensity of the cell at 0 s time point by the average intensity before bleaching. An average Max recovery value was obtained from all the bleached cells in the experiment.

To correct for photobleaching during the experiment, a different set of spots was manually selected in cells not exposed to localized bleaching, so they could serve as a baseline control. An average bleaching curve was produced using the intensity traces from these fluorescent foci. All data used to generate the bleaching curve were obtained in the same day using the same strain, excitation settings and interval between pictures as for the FRAP experiment. The average curve was fitted to an exponential decay function. FRAP intensity traces were corrected by dividing each time point by the corresponding normalized value in the fitted bleaching curve.

Data from the same set of experiments were averaged. Data from experiments performed the same day, but having different intervals between pictures, were collated into a single recovery curve. Data were then fitted by an exponential solution of the reaction-diffusion equation in a reaction-limited regime (*equation 1*) using MATLAB:

$$y = c - ae^{-bt} \tag{1}$$

where $c$ is the asymptote for recovery, $a$ the amplitude of recovery, and $b$ the rate of unbinding (i.e. $k_{off}$). Bound-times are the reciprocal of $k_{off}$.

Upper boundary for $c$ during fitting was set to the Max recovery (see above), plus ten percent of this value to account for measurement error. In addition to R squared, which is not recommended for non-linear models, goodness of fit was assessed using the Kolmogorov-Smirnov test by measuring the normality in the distribution of residuals (*Andrae et al., 2010*). Standard errors and 95% confidence intervals (CI) on the parameter estimates were calculated using the variable values previously obtained, as initial estimates, and bootstrap sampling was performed over 10,000 samples (*Supplementary file 1A*). The values reported in the figures are weighted averages of all the experiments done for the same subunit.

We expect that co-localization of sister replisome will have no effect on the rates calculated since the intensity of every spot is normalized against itself in FRAP, and the average rate of recovery is the same at every replisome. Similarly, in sptPALM binding time of individual molecules should not be influenced by a nearby replisome, resulting only a minimal increase in the probability of re-binding to the same place.

## sptPALM

Cells were harvested from early log-phase cultures in M9-Glycerol (OD$_{600}$0.1–0.2), concentrated and spotted onto a pad of 1% agarose in M9-Glycerol, contained in a gene frame. Coverslips cleaned with versa-clean, acetone and methanol were used to minimize fluorescent background. Treatment with hydroxyurea was done on the agarose pad, by mixing HU with media and agarose.

Imaging was performed at room temperature on an inverted Olympus IX83 microscope using a 60x oil objective lens (Olympus Plan Apo 60X NA 1.42 oil) or 100x oil objective lens (Olympus Plan Apo 100X NA 1.40 oil). Images were captured using a Hamamatsu Orca-Flash 4.0 sCMOS camera. Excitation was done from an iChrome Multi-Laser Engine from Toptica Photonics. Laser triggering was done through a real-time controller U-RTCE (Olympus). Experiments were done using HiLo illumination setup (*Tokunaga et al., 2008*) from a single-line cellTIRF illuminator (Olympus). Olympus CellSens 2.1 imaging software was used to control the microscope and lasers.

For experiments with replisome subunits fused to mMaple, a single 405 nm wavelength activation event, typically lasting less than 20 ms, was followed by multiple 561 nm wavelength excitation events with camera captures of 500 ms spaced by 1 s or 5 s intervals, or camera captures of 2s with continuous excitation (2 s rates) or 10 s intervals. Low levels of exposure to violet-blue light were used to minimize photoxicity and allow cells to continue growing during the experiments (*Figure 1—figure supplement 3*). To image LacI, we used continuous illumination of 561 nm wavelength after a

single 405 nm wavelength activation event at capture rates and intervals of 500 ms or 2 s. We also used 2s capture with 10 s intervals to characterize LacI bleaching in long experiments. Rifampicin experiments were done in a similar manner except Rifampicin was added to the M9-Gly agarose pad, and imaging was done after a 20 min incubation on the agarose pad. We noticed that fewer spots were detected, consistent with inhibition of replication initiation through Rifampicin.

## sptPALM analysis

Images were first segmented in order to remove out-of-cell noise coming from contaminants on the coverslip. Binary masks were created using ImageJ, either from the differential interference contrast (DIC) channel or the green fluorescent channel of mMaple. For DIC, alignment was done by first obtaining a maximum-intensity projection of the PALM timelapse, and subsequently aligning it to the reference DIC, using ImageJ. Each slice of the PALM timelapse was then multiplied by the binary mask, to retain intensities within cells only. An average value of the out-of-cell background was added to regions outside of the ROIs to minimize incorrect assignment by the detection program due to sharp intensity increases.

PALM tracking was performed using previously developed software (*Uphoff et al., 2013*), based on the DAOSTORM (*Holden et al., 2011*) localization algorithm. An intensity threshold was used to find candidate molecules. The positions of the candidate molecules were then used as initial guesses for a 2D-elliptical Gaussian fit. The fitted parameters were: x-position, y-position, x-standard deviation, y-standard deviation, intensity, brightness, elliptical rotation angle, and background. Tracking was done based on a widely used algorithm (*Crocker and Grier, 1996*). Localizations were linked if they appeared within a 300 nm radius between consecutive frames, using a memory parameter of one frame to account for blinking or missed localizations (i.e. the molecule can go missing for one frame and still be linked).

Further refinement of the recorded tracks was done to analyze only those that represented immobile single-molecules. To remove slow-diffusing molecules, we plotted a histogram of the PSFs in x and y for all localizations, and performed a two-component Gaussian mixture fit using Maximum Likelihood Estimation. The component with the smaller mean PSF likely represents bound molecules, whereas the other component represents unbound molecules. The two-component Gaussian mixture model has the following form:

$$p\left(\frac{1}{\sigma_1\sqrt{2\pi}}\right)e^{\frac{-(x-\mu_1)^2}{2(\sigma_1)^2}} + (1-p)\left(\frac{1}{\sigma_2\sqrt{2\pi}}\right)e^{\frac{-(x-\mu_2)^2}{2(\sigma_2)^2}} \qquad (2)$$

Where $p$ is the mixture probability, $\sigma_1$ and $\mu_1$ are the standard deviation and mean of normal distribution 1, respectively. Likewise, $\sigma_2$ and $\mu_2$ are the standard deviation and mean of normal distribution 2, respectively. From the fit, we identified the mean and standard deviation of the component representing bound molecules. We then took 2 standard deviations above the mean to obtain an initial estimation of the threshold. We assessed the accuracy of tracking by manually comparing the tracking results for a subset of fluorescent spots with their lifetime in the original images. Using this method we determined that a threshold of $x \leq 170$ and $y \leq 215$, placed on the mean PSF over the track, helped to eliminate most of the unbound molecules from subsequent analysis.

In addition, we varied the threshold on the number of localizations for track acceptance across different time-intervals of capture. Our reasoning was that the probability that a track represents a genuinely bound molecule becomes higher as the time interval used increases (*Mazza et al., 2012*). Therefore, the thresholds for removing tracks were <4, <3, <2, and <2 localizations for interval times of 1 s, 2 s, 5 s, and 10 s, respectively. The thresholds were selected by comparing the raw image by eye to the tracks found by the tracking software. Technically no tracks were removed for 5 s and 10 s since tracks with 1 localization cannot be used to calculate track durations.

To quantify only single-molecule tracks, we plotted a histogram of the mean intensity of a track and fit using a Gaussian Mixture Model (GMM), utilizing the Expectation-Maximization (EM) algorithm. The intensity values were clustered based on membership probabilities (i.e. the probability of belonging to a particular Gaussian component). We used a 2 component GMM fit for most cases and isolated the cluster having the lowest mean, as the intensity values from this cluster likely represent single molecules. We used a 3 component GMM fit in some cases where a significant portion of the molecules seemed out of focus, resulting in a sharp spike of low intensity values in the

histogram. In such cases, we isolated the cluster with the second lowest mean. This was especially important when studying proteins with long bound-times, where track fragmentation has a greater relative effect in underestimating the real track duration. We also performed a Bayesian Information Criterion (BIC) test to confirm that the 3 component GMM fit better than the 2 component model. We used only track durations with single molecule intensity values for subsequent analysis.

To avoid track fragmentation in the analysis of proteins with long bound-times, as in LacI and DnaB, caused by fluctuations in intensity or the molecule moving transiently out-of-focus, we determined the typical length of time that the localization software misses spot detection during long tracks (gap time). We did this by manually comparing the outcome of the analysis to the lifetime of a subset of spots in the original images. We found that on average, the gap was ~4 frames. Therefore, we linked tracks based on the criteria that their mean positions were $\leq$300 nm apart and gap time between them was <= 4 frames. For these data sets, we performed the GMM fit for isolation of single-molecule tracks after track linkage.

The track durations of multiple samples taken on the same day and time-interval were amalgamated into one data set. In order to get the average track duration, we fitted the track durations using MLE. The reason for our choice of MLE over the more commonly used Least Squares-Estimation (LSE) method is that it is invariant to the bin size (i.e. the parameter estimate is the same regardless of how we bin the data) and it allows us to infer what the population parameter is. Essentially, we use information from our sample data (track duration times and track acceptance threshold) as input into MLE, in order to find the population probability density function (PDF), that makes our data the most likely. The fitted lines represent this PDF (*Myung, 2003*; *Woody et al., 2016*).

Histograms were binned based on the square-root rule, where the number of bins is equal to the square root of the sample size. We binned our data for presentation purposes only, in order to reduce noise associated with a finite sample size and reveal our sample distribution more clearly.

The PDF of the track durations is related to bleaching and unbinding as follows:

$$k_{track}e^{-k_{track}t} = \left(k_{off} + k_{bleach}\right)e^{-\left(k_{off}+k_{bleach}\right)t} \tag{3}$$

Where $k_{track}$ is the rate of track durations ending, $k_{off}$ is the rate of unbinding, and $k_{bleach}$ is the rate of bleaching. The model PDF we used for fitting was a left-truncated exponential distribution. This was used to compensate for the fact that we removed short duration tracks from analysis. The general form of this PDF is:

$$\left(\frac{1}{\tau}\right)e^{\frac{-(x-L)}{\tau}} \tag{4}$$

where $\tau$ is the mean time, and $L$ is the truncation point/origin of exponential distribution (*Balakrishnan and Basu, 1995*). Note that the equation has the same form as expected for a translated exponential distribution and so we used this form for all data sets.

To correct for photobleaching, we used LacI tagged with mMaple. LacI is expected to have a binding time significantly longer than the bleaching time of mMaple in our experimental conditions (*Hammar et al., 2014*). Therefore, since the $k_{off}$ term is much smaller than the $k_{bleach}$ term, the average track duration is equivalent to the average bleach time for mMaple. Note that previous estimates of LacI bound-time at the *lacO* operator were determined using a single copy of the operator, while we used an array composed of 256 copies of *lacO*. This should result in even longer apparent binding of the repressor protein and increase the likelihood that focus disappearance is solely due to bleaching. We obtained a constant exposure bleaching curve, which we used for the 1 s, and 5 s intervals (500 ms exposure data). We scaled the average bleach time from the constant exposure bleaching data, in order to use it for different time intervals. The constant exposure bleaching time is related to the average bleaching time as follows:

$$T_{bleach} = \left(\frac{t_{interval}}{t_{exp}}\right)T_{constant} \tag{5}$$

Where $t_{interval}$ is the interval time, $t_{exp}$ is the exposure time, and $T_{constant}$ is the constant exposure bleaching time.

We then calculated the average bound time using the following equation:

$$T_{bound} = \frac{T_{track}T_{bleach}}{T_{bleach} - T_{track}} \tag{6}$$

In order to calculate the SE and 95% CI on the parameter estimates, we used the right-hand side of *Equation 3*. We used the bleaching times and bound times calculated previously, as initial estimates, and then performed bootstrap sampling over 10,000 samples, in order to calculate the standard errors and confidence intervals on the bound time estimates. We used the 'bias and accelerated percentile method' (BCA) algorithm when calculating CI, in order to compensate for any bias or skewness in the bootstrap distribution.

Previous characterization of photoblinking of mMaple found 49% probability of blinking and an average of 3.4 blinking events per molecule (*Durisic et al., 2014*). This same study set a cutoff time of 2.6 s to account for over 99% of the blinking events. We expected to detect fewer blinking events since the shorter ones will be recorded only as intensity fluctuations, and not discontinuities in the track, due to the use of longer capture rates, 500 ms instead of 100 ms. In addition, lower exposure intensity would likely contribute to a decrease rate of blinking (*Garcia-Parajo et al., 2000*). To estimate the effect of blinking in our analysis, we used the data of LacI using 500 ms capture-times. We analysed the data as previously except that we did not apply the one-frame memory parameter during tracking. We then determined the number of frames between two consecutive tracks at the same position of the field of view. We used a 2.6 s cutoff in our data since longer gap times likely represent new binding events instead of blinking.

For DnaB, since 500 ms capture-times were not efficient at preventing diffusing molecules from being detected, even after the PSF threshold was applied (*Figure 2—figure supplement 4G*), we used exposure times of 2 s and spaced capture by intervals of 2 s and 10 s. We used Pol III ε subunit as a control to ensure that increasing the exposure time does not significantly alter the bound time estimates (*Figure 2—figure supplement 3*).

Since the track durations of DnaB are similar to those of LacI, we determined a weighted average of the track duration times obtained and for each data set of DnaB performed a constrained fit (i.e. fitting with bounds placed on the estimates). We calculated a bound time from the weighted average in order to generate an initial estimate of the bound time, which was then used for the constrained fit. We allowed for 20% variation in the bleaching time in order to determine physically reasonable estimates. The lower and upper bounds for the DnaB bound time were 1 s and 90 min, respectively.

For the fitting procedure, we calculated the negative log-likelihood function of the two parameter (bleach time, bound time) left-truncated exponential distribution, as well as the gradients. We then used the MATLAB minimization function, *fmincon*, in order to find the parameters that minimize the negative log-likelihood function. This was done to improve the convergence to the correct solution, especially if the initial estimates were far from the actual solution, and to simplify the estimation procedure. We subsequently performed bootstrap sampling as discussed before to calculate standard errors and confidence intervals (*Supplementary file 1B*).

The final estimates for bound times were calculated by doing a weighted average of data taken on multiple days and with different time intervals.

We performed a chi-square goodness of fit test under the null hypothesis that our data is sampled from a single-exponential distribution, and the alternative hypothesis that it does not. It is possible however that even if the fit is good, that a different model fits the data better (e.g. two exponential model). We wanted to determine the best model for the data and we performed two tests in this regard: (1) Log-Likelihood Ratio(LLR) test and (2) Bayesian Information Criterion (BIC) test.

Log-Likelihood Ratio Test- The LLR test tries to test if an unconstrained model statistically significantly fits the data better than the constrained model, by comparing the likelihood values obtained from the unconstrained versus the constrained. In our case, the unconstrained model is the two-exponential model while the one-exponential model is the constrained model, as shown below:

$$p\left(\frac{1}{\tau_1}\right)e^{\frac{-(x-L)}{\tau_1}} + (1-p)\left(\frac{1}{\tau_2}\right)e^{\frac{-(x-L)}{\tau_2}} \tag{7}$$

where $\tau_1 = (T_{bleach} + T_{bound\alpha})/T_{bleach} * T_{bound\alpha}$, $\tau_2 = (T_{bleach} + T_{bound\beta})/T_{bleach} * T_{bound\beta}$, p is the mixture probability, and L is the truncation point.

Note that if we constrain p=1, we recover the single-exponential model.

Bayesian Information Criterion (BIC) test- The BIC test determines which model fits the data better, but penalizes for greater complexity (i.e. more parameters), to prevent over-fitting the data. The lowest number obtained through the test indicates the model that fits the data the best with the least complexity.

When calculating the MLE estimates and log-likelihood values for the two-exponential model, the lower and upper bounds for the two timescales were 0.1 and 5400 s, respectively. The bounds for the bleaching constant were placed such that it allowed for 20% variation in the estimate.

The criterion we used to judge if the two-exponential model was the better model, was if the BIC test gave the lowest value for the two-exponential model and the LLR test gave a p<0.01. Also, the estimates obtained from the two-exponential should be sensible, and especially, they should not give us simply the values of the bounds, as that indicates that no estimates were found.

We found a few cases where the dataset passed the criterion. The timescales estimated were not consistent however, and upon further examination we realized that it was due to a few noticeable outliers in the dataset, possibly from noise due to dirt still on the coverslip. When we removed the outliers, it resulted in these datasets not passing the criterion, but without significantly changing the bound times previously obtained in the single-exponential fits.

## Blinking analysis of mMaple for sptPALM

We used LacI data collected with 500 ms exposure as fast as possible (~500 ms interval time), to characterize mMaple under our acquisition settings. The mean positions of single-molecule tracks initiating at the first frame were used as ROIs around which a $7 \times 7$ pixel window was drawn to extract intensity-time traces. Fluorescence from a bound molecule was identified as being 2 standard deviations above the mean cellular background, and the signal had to be above this threshold for >3 localizations (similar to track acceptance threshold previously described). Gap durations were calculated as the number of frames between bound fluorescence signals. Fits to the gap durations were done through MLE using a truncated exponential model. The resulting fit was used to calculate the probability of a gap duration lasting greater than a specified value, through integration.

## Estimation of β-clamp loading rate

To estimate the effective loading rate we followed the following equation described elsewhere (*Moolman et al., 2014*):

$$\frac{1}{T_{eff\_load}} = \frac{\beta_{2bound}}{T_{unload}} \tag{8}$$

where $T_{eff\_Load}$ represents the loading rate, $\beta_{2bound}$ represents the number of copies of $\beta$ clamp at the fork and $T_{unload}$ represents the bound-time of a $\beta$ clamp. In our estimations we assume that there are 23 $\beta$ dimers per fork as previously estimated (*Moolman et al., 2014*).

## Acknowledgements

We thank Ann McEvoy for kindly providing a plasmid carrying mMaple. We thank David Sherratt, Adam Hendricks, James Graham, Jackie Vogel, Daniel Jarosz, Charl Moolman and members of the Reyes-Lamothe lab for discussion and helpful comments on this work. SU was supported by a Sir Henry Wellcome Fellowship by the Wellcome Trust and a Junior Research Fellowship at St John's College Oxford. AJMW and MCL were funded by the Medical Research Council UK (MRC# MR/K01580X/1), the Biology and Biotechnology Research Council UK (BBSRC# BB/N006453/1) and the Biological Physical Sciences Institute (BPSI) at the University of York, UK. This work was funded by the Natural Sciences and Engineering Research Council of Canada (NSERC# 435521–2013), the Canadian Institutes for Health Research (CIHR MOP# 142473), the Canada Foundation for Innovation (CFI# 228994), and the Canada Research Chairs program.

## Additional information

### Funding

| Funder | Grant reference number | Author |
|---|---|---|
| Natural Sciences and Engineering Research Council of Canada | Discovery Grant,435521-2013 | Thomas R Beattie<br>Nitin Kapadia<br>Rodrigo Reyes-Lamothe |
| Canada Research Chairs | Tier II,950-228994 | Rodrigo Reyes-Lamothe |
| Canadian Institutes of Health Research | Operating Grant,142473 | Thomas R Beattie<br>Nitin Kapadia<br>Rodrigo Reyes-Lamothe |
| Canada Foundation for Innovation | Leaders Oportunity Fund,228994 | Thomas R Beattie<br>Nitin Kapadia<br>Rodrigo Reyes-Lamothe |
| Wellcome | Junior Research Fellowship | Stephan Uphoff |
| Biotechnology and Biological Sciences Research Council | BBSRC# BB/N006453/1 | Adam JM Wollman<br>Mark C Leake |
| Medical Research Council | MRC# MR/K01580X/1 | Adam JM Wollman<br>Mark C Leake |

The funders had no role in study design, data collection and interpretation, or the decision to submit the work for publication.

### Author contributions

TRB, NK, Formal analysis, Investigation, Writing—original draft, Writing—review and editing; EN, Investigation, Writing—original draft; SU, Software, Investigation, Writing—original draft, Writing—review and editing; AJMW, Software, Writing—original draft, Writing—review and editing; MCL, Formal analysis, Writing—original draft, Writing—review and editing; RR-L, Conceptualization, Supervision, Funding acquisition, Investigation, Writing—original draft, Writing—review and editing

### Author ORCIDs

Nitin Kapadia, http://orcid.org/0000-0003-4514-7631
Rodrigo Reyes-Lamothe, http://orcid.org/0000-0002-5330-3481

## Additional files

### Supplementary files

• Suplementary file 1. Summary of the analysis for FRAP and sptPALM. (A) Analysis of FRAP data. (B) Analysis of sptPALM data. (C) Results of goodness-of-fit tests for sptPALM.

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
