## [Decision Letter]

Thank you for submitting your article "Frequent exchange of the DNA polymerase during bacterial chromosome replication" for consideration by *eLife*. Your article has been reviewed by two peer reviewers, and the evaluation has been overseen by a Reviewing Editor and Jessica Tyler as the Senior Editor. The reviewers have discussed the reviews with one another and the Reviewing Editor has drafted this decision to help you prepare a revised submission. The reviewers have opted to remain anonymous.

Overall, the referees are in agreement that the work potentially represents an important advance in understanding the dynamics of the bacterial replisome in vivo. However, the referees also raised several questions regarding the experimental methods used and data interpretation. Before a decision can be made regarding publication, we ask that you submit a revised manuscript that addresses the comments below.

Essential Changes:

1) The model that the holoenzyme dissociates from DnaB as a single unit is based on the observation that tau, epsilon, α and δ have identical timescales of dissociation. There are large uncertainties in the DNA bound times and the data in its current form are not strong enough to make this point. In this regard, the authors should provide good characterizations of the fluorescence properties of Ypet and mMaple. Most fluorescent proteins blink (switching reversibly between a bright and dark state), and mMaple is in particular known for its blinking property (Wang, PNAS, 2014). Fluorescence recovery after photobleaching could be spontaneous instead of due to subunit exchange (which could explain the strange behavior of DnaB FRAP experiment – DnaB needed 2s exposure time to prevent diffusing molecules from detected, and hence the authors assign the ~7sec initial FRAP recovery for DnaB to diffusing molecules. Yet single molecule tracking showed DnaB diffuses much slower than that). The authors should conduct control experiments so that these effects could be subtracted (for example using fixed cells or no-exchanging proteins under all conditions used in the paper).

2) In general, one replication fork has two copies of active Pol III. If the two replication forks stay with each other that will give a maximum of four copies. Even with three copies of Pol III as the corresponding author previously published (Reyes-Lamothe, Science, 2012), there are only maximally six labeled copies for each subunit. As such, should the authors observe a stepwise increase of single molecule fluorescence after photobleaching (as in the previous paper) if there are new subunits coming in? The authors observed gradual fluorescence increase instead, which also raises the possibility that this could be due to spontaneous recovery of fluorescence as mentioned in point 2. Comment on point is needed in the text/Methods as appropriate.

3) Similar concern goes to the rebinding experiments. Can the authors conduct control experiments to show that this is not complicated by the blinking behaviors of mMaple?

4) The track duration fits in Figure 2 do not appear to fit the data very well and often only span a few bins. Related to this point, how did the authors determine the appropriate bin sizes to use in the histograms? In some cases, it seems as though relatively few bins were used to generate the distributions. For example, the distributions in Figure 2—figure supplement 3 are plotted with only 5 or 6 bins although (according to the values in [Supplementary-material SD1-data]) they seem to represent several hundred molecules.

5) It is unclear how results from multiple experiments were combined. For example, the bound time for ε in Figure 2—figure supplement 3 for the 500 ms exposure seems to be the value given in [Supplementary-material SD1-data] for the second of the three experiments listed under that condition. Likewise, the bound time for the 2s exposure seems to be from the second of the two experiments under that condition. Are the values given in the text and the main figures calculated from a single experiment?

6) The authors show that the bound lifetimes for components of the holoenzyme are much shorter than the lifetime of DnaB. The authors ascribe this difference to the holoenzyme dissociating frequently from DnaB. How do the authors know that the dynamics of the holoenzyme are not dominated by associations with the large number of clamps behind the replication fork? The same question applies to Figure 3 where the authors assume that DnaB is the platform for rebinding, alternatively epsilon could be associating with clamps near the replisome.

7) It is interesting that Pol III still turns over when there is no DNA synthesis. The authors proposed that frequent encounters with transcription or DNA binding proteins result the turnover. The authors may wish to inhibit transcription to test the hypothesis.

8) In the Discussion, the authors invoke dynamic processivity as a mechanism that could explain the relatively transient binding of the holoenzyme. Such models require that the competitor is at a high local concentration which is often generated by nearby binding. It is hard to imagine that a second holoenzyme is associating with DnaB prior to dissociation of the active holoenzyme. It seems a more likely model is that an obstacle that impedes Pol III triggers a conformational transition resulting in a decrease in affinity and release. Such models have been used to describe the recycling of Pol III upon collision with an Okazaki fragment.

9) For the FRAP experiment, Cephalexin treated cells were used and assumed to have the same replisome dynamics as untreated cells, this is mostly based on the similar ori1/SSB ratio (Figure 1—figure supplement 1). The increase in cell volume with drug treatment could potentially influence replisome dynamics (like that the authors later pointed out protein concentration matters). Although life-time of replisome components are similar when probed with sptPALM as described later in the article, it would be good to independently check if DNA/RNA synthesis is effected by Cephalexin perhaps via synthesis rate measurements, or at least provide control experiments to show how this looks like in WT, untreated cells.

10) A significant concern with the authors' model is that it is unclear why, if the holoenzyme were indeed dissociating so frequently, there wouldn't be a much higher copy number of holoenzyme components behind the fork. Leake and Reyes-Lamothe have previously argued for only three copies of pol III within replication foci. This leads to the question of which polymerase is filling in all of these gaps?

11) A general point regarding the Discussion is that the authors provide no potential alternative interpretations of their data.

Suggested changes:

1) In *E. coli*, the two replisomes from two forks overlap with each other for a significant portion of time (such as in Figure 1—figure supplement 2 panel C. 30min to 40 min to 50 min). All analyses are done under the assumption that a large fluorescence spot could contain either one or two replisomes (cannot be resolved visually due to short distance separation). Perhaps the authors should state this more explicitly in the text and analyses.

2) In discussing the FRAP results for DnaB, the authors attribute the incomplete fluorescence recovery to slow diffusion of DnaB. This logic isn't clear and should be explained in greater detail.

3) Figure 1 and Figure 4 have three y-axes, and are difficult to comprehend. Separating all the measurements, or using broken bars will be easier for the readers.

4) The data plotted in Figure 2 might be better represented in a table. The bar graph doesn't add anything and could be slightly confusing because the y axes are scaled differently.

5) The authors mentioned using a reaction-diffusion model to analyze FRAP traces. There was no description anywhere in the manuscript about what the model is. It appears that the authors simply used an exponential fitting and extracted the half time. If so please state clearly and do not use the name of reaction-diffusion, which has a specific meaning.

6) There are typos in Figure 2 and Figure 2—figure supplement 3 ("Bleach time" legend).

7) Please provide n values (sample size) for all statistic measurements with error bars.

8) Regarding additional data files and statistical comments – the authors should provide n values for all measurements.

[Editors' note: further revisions were requested prior to acceptance, as described below.]

Thank you for resubmitting your work entitled "Frequent exchange of the DNA polymerase during bacterial chromosome replication" for further consideration at *eLife*. Your revised article has been favorably evaluated by Jessica Tyler (Senior Editor), a Reviewing Editor, and two reviewers.

The manuscript has been improved but there are a few remaining issues that need to be addressed before acceptance, as outlined below:

Primary points:

1) Some concerns about the quality of the lifetime fits for the PALM data still exist. From the presentation of the data in the figures, it's very hard to assess the quality of the fits, and many of the conclusions of this paper (exchange of the entire Pol III holoenzyme, no evidence of distinct leading and lagging-strand polymerase dynamics, etc.) rely heavily on those lifetimes. The response raised in the previous version about the presentation of data in histograms and the number of bins used is not wholly clear. For example, the epsilon lifetime data in Figure 2—figure supplement 3 appear to include 143 molecules for the 500 ms exposure and 415 molecules for the 2s exposure. Yet the histograms appear to contain data in only 5 or 6 bins, which is not consistent with the square root of N binning claimed to be used. Even if the MLE fitting method is insensitive to the choice of bins, the current presentation of the data makes it difficult to see how well the data are fit by the resulting curves. Please resolve this issue.

2) Regardless of the exact timescales of exchange, it has not been shown that the timescales of the holoenzyme components are identical. The Discussion should be amended to discuss different scenarios for replisome dynamics, such as the possibility that individual components of the holoenzyme exchange relatively rapidly with free proteins in solution without the complex falling apart. The manuscript attempts to argue against this option by noting that epsilon dissociation and rebinding is observed more frequently than one would anticipate from the free pool of epsilon. This argument is tenuous, however, as no direct evidence is provided to show that these epsilon molecules are diffusing with the holoenzyme (to test this, for example, one could determine whether those epsilon molecules that dissociate and rebind have an anomalously slow diffusion constant consistent with the much larger holoenzyme). Please amend the discussion of the model to point out that the favored model is at best suggested by the data, and please also present some alternative interpretations.

Secondary comments:

1) Regarding the possibility of two timescales in the lifetime data, it would help the reader assess the quality of these fits by plotting the data on a semi-log axis, so that deviations from linearity are more apparent.

2) Concerning the section "DnaB is a stable platform upon which the PolIII holoenzyme exchanges": Measurement of DnaB's long bound lifetime demonstrates that it is a stable component of the replisome, but not necessarily a platform upon which PolII assembles; this latter claim is likely based on what is known about the replisome, by not experimentally tested here (e.g., a DnaB mutant that fails to interact with tau would need to be used to examine this hypothesis). The section title could be altered slightly to more accurately reflect the experimental results.

3) Figure 2—figure supplement 4. The color of DnaB and epsilon appears to be switched. In panels E and F, there appeared to be negative bins of the apparent diffusion coefficients? Please fix.

---

## [Author Response]

*Essential Changes:*

*1) The model that the holoenzyme dissociates from DnaB as a single unit is based on the observation that tau, epsilon, α and δ have identical timescales of dissociation. There are large uncertainties in the DNA bound times and the data in its current form are not strong enough to make this point. In this regard, the authors should provide good characterizations of the fluorescence properties of Ypet and mMaple. Most fluorescent proteins blink (switching reversibly between a bright and dark state), and mMaple is in particular known for its blinking property (Wang, PNAS, 2014). Fluorescence recovery after photobleaching could be spontaneous instead of due to subunit exchange (which could explain the strange behavior of DnaB FRAP experiment – DnaB needed 2s exposure time to prevent diffusing molecules from detected, and hence the authors assign the ~7sec initial FRAP recovery for DnaB to diffusing molecules. Yet single molecule tracking showed DnaB diffuses much slower than that). The authors should conduct control experiments so that these effects could be subtracted (for example using fixed cells or no-exchanging proteins under all conditions used in the paper).*

We are now including a new figure, Figure 1—figure supplement 2, where we show that there is less than 5% recovery after bleaching in fixed cells carrying YPet, consistent with stochastic fluctuations and experimental measurement error. We therefore estimate that the contribution of blinking to recovery after photobleaching is minimal in our experiments.

Note that further support for this conclusion is provided by: 1) similar bound-time estimates for the subunits studied in both FRAP and sptPALM; 2) no observed recovery, qualitatively, for DnaB-YPet (Figure 1); 3) much slower bound-times estimated in other non-replisome proteins using a similar approach (Badrinarayanan 2012; MukB bound time estimated to be ~50seconds).

*2) In general, one replication fork has two copies of active Pol III. If the two replication forks stay with each other that will give a maximum of four copies. Even with three copies of Pol III as the corresponding author previously published (Reyes-Lamothe, Science, 2012), there are only maximally six labeled copies for each subunit. As such, should the authors observe a stepwise increase of single molecule fluorescence after photobleaching (as in the previous paper) if there are new subunits coming in? The authors observed gradual fluorescence increase instead, which also raises the possibility that this could be due to spontaneous recovery of fluorescence as mentioned in point 2. Comment on point is needed in the text/Methods as appropriate.*

As commonly done in the analysis of FRAP data (McNally JG, Methods Cell Biol, 2008, 85:329), the traces were first averaged – in order to reduce the noise inherently to this type of experiments – and the model was fit onto the resulting average curve (please see the Methods section). Recovery of fluorescence in the average curve is gradual, but in individual traces there are often much greater changes between time points, supporting the idea that multiple subunits bind simultaneously.

*3) Similar concern goes to the rebinding experiments. Can the authors conduct control experiments to show that this is not complicated by the blinking behaviors of mMaple?*

We are now including a new figure, Figure 2—figure supplement 3, where we show a quantitative characterization of mMaple blinking under our experimental conditions. Overall, we do not find evidence that blinking has a significant effect in our experiments.

Note that the use of longer exposure times (500ms) results in the need of much lower excitation light intensity when compared to other single-molecule studies (typically using capture rates of 15-20ms). Since the probability to induce blinking is linked to the intensity power used (Garcia-Parajo, 2000, PNAS, 97:7237), we expect that the intensity level used in our experiments induce blinking less frequently than in other studies. Furthermore, the use of long capture rates also means that short blinking events will only be detected as fluctuations in the intensity of the spots, as blinking will be averaged out over the exposure time.

The use of a “memory parameter” during tracking, allowing for the disappearance of spot for a single frame without segmentation of the track, is sufficient to eliminate most of the effects of blinking. We estimate that our analysis will prematurely terminate less than 7.5%, 3% and 0.0001% of the tracks due to blinking when using a 1-second, 2-second, and 5-second intervals, respectively.

We also note that if blinking had a significant effect on our results, it would have resulted in similar bound times across the different proteins, as they share the same local environments and therefore the photophysics of mMaple would be the same.

*4) The track duration fits in Figure 2 do not appear to fit the data very well and often only span a few bins. Related to this point, how did the authors determine the appropriate bin sizes to use in the histograms? In some cases, it seems as though relatively few bins were used to generate the distributions. For example, the distributions in Figure 2—figure supplement 3 are plotted with only 5 or 6 bins although (according to the values in [Supplementary-material SD1-data]) they seem to represent several hundred molecules.*

We binned our data using the commonly used Square-Root rule where the number of bins is equal to the square root of the sample size. However, the estimates obtained through MLE are invariant to the bin size (i.e. we will get the same parameter estimate regardless of how we bin the data). This in contrast to the Least-Squares Estimation (LSE) method where the estimates obtained are dependent on the bin size.

Another reason why we chose MLE over LSE for the estimation of bound-times is that it allows us to infer the population from which our sample data came from. In this sense, we are using information from our data (e.g. track duration times and threshold for track acceptance) as input into MLE, in order to obtain an estimate of the population parameter. The fitted lines shown represent the population probability density function that makes the sample data the most likely.

We still bin our data, but only for presentation purposes, in order to reduce noise associated with a finite sample size, and reveal our sample distribution more clearly.

We have clarified this point in the Methods section.

*5) It is unclear how results from multiple experiments were combined. For example, the bound time for ε in Figure 2—figure supplement 3 for the 500 ms exposure seems to be the value given in [Supplementary-material SD1-data] for the second of the three experiments listed under that condition. Likewise, the bound time for the 2s exposure seems to be from the second of the two experiments under that condition. Are the values given in the text and the main figures calculated from a single experiment?*

Figure legend in Figure 2—figure supplement 3 has been corrected. It is now clear that the data comes from a single set of experiments.

Description on how multiple experiments were combined is included in the Methods section. In short, for FRAP a single average curve was obtained from all the experiments – including different intervals between captures – of a single day for a particular replisome subunit. The model was fitted into this curve to obtain the estimated bound-times. Results from multiple days were combined using weighted averages.

For sptPALM, the data for a single set of experiments using the same time interval were combined and the bound-time was estimated. Estimated bound-times from different days and using different intervals were combined using weighted averages.

*6) The authors show that the bound lifetimes for components of the holoenzyme are much shorter than the lifetime of DnaB. The authors ascribe this difference to the holoenzyme dissociating frequently from DnaB. How do the authors know that the dynamics of the holoenzyme are not dominated by associations with the large number of clamps behind the replication fork? The same question applies to Figure 3 where the authors assume that DnaB is the platform for rebinding, alternatively epsilon could be associating with clamps near the replisome.*

We tested for the presence of two exponentials in our data – which would reveal if there were two different populations of holoenzyme binding to DnaB and the clamps – but did not find any evidence for them. Analysis of the data by eye does not show any clear evidence for two populations either. The stoichiometry of the polymerase and the clamp loader at the replication fork, determined in an earlier paper and replicated by other groups, does not agree with multiple copies of the replisome per fork, so if there are spare subunits binding to β clamp, we expect that they will not overwhelm the dynamics at the fork. Furthermore, we do not observe evidence for bound times of Pol III at ~40s, similar to that of β clamp.

*7) It is interesting that Pol III still turns over when there is no DNA synthesis. The authors proposed that frequent encounters with transcription or DNA binding proteins result the turnover. The authors may wish to inhibit transcription to test the hypothesis.*

We have repeated the experiments with epsilon in the presence of rifampicin (Figure 4). The results show that the presence of this drug did not significantly change the rate of turnover of epsilon when compared to the untreated sample. We conclude that transcription-DNA replication collisions do not account for the frequent unbinding of the polymerase.

*8) In the Discussion, the authors invoke dynamic processivity as a mechanism that could explain the relatively transient binding of the holoenzyme. Such models require that the competitor is at a high local concentration which is often generated by nearby binding. It is hard to imagine that a second holoenzyme is associating with DnaB prior to dissociation of the active holoenzyme. It seems a more likely model is that an obstacle that impedes Pol III triggers a conformational transition resulting in a decrease in affinity and release. Such models have been used to describe the recycling of Pol III upon collision with an Okazaki fragment.*

We have extended the Discussion to include alternative models that explain our data.

*9) For the FRAP experiment, Cephalexin treated cells were used and assumed to have the same replisome dynamics as untreated cells, this is mostly based on the similar ori1/SSB ratio (Figure 1—figure supplement 1). The increase in cell volume with drug treatment could potentially influence replisome dynamics (like that the authors later pointed out protein concentration matters). Although life-time of replisome components are similar when probed with sptPALM as described later in the article, it would be good to independently check if DNA/RNA synthesis is effected by Cephalexin perhaps via synthesis rate measurements, or at least provide control experiments to show how this looks like in WT, untreated cells.*

We are now including more data in Figure 1—figure supplement 1 that further supports our claim that cephalexin does not have an effect on DNA or protein synthesis. Note that even though longer cells have a greater copy number for any given protein, we expect that the concentration of the protein will be very similar to that in untreated cells.

*10) A significant concern with the authors' model is that it is unclear why, if the holoenzyme were indeed dissociating so frequently, there wouldn't be a much higher copy number of holoenzyme components behind the fork. Leake and Reyes-Lamothe have previously argued for only three copies of pol III within replication foci. This leads to the question of which polymerase is filling in all of these gaps?*

Currently we can only speculate on what happens to DNA on the leading strand after Pol III holoenzyme detaches. Note that the binding kinetics we observe suggest that Pol III HE unbinds from both helicase and DNA. Currently our favored model is that detachment of Pol III HE from DnaB will result in a ~10 slower progression of the helicase (see third paragraph in the Discussion section). If recruitment of a new Pol III HE occurs shortly after (or simultaneously with the removal of the previous copy), then it can in principle use the 3’ end (still bound to β clamp) and leave no gap at the leading strand. Alternatively, it may require priming and gap filling.

*11) A general point regarding the Discussion is that the authors provide no potential alternative interpretations of their data.*

We have now extended the Discussion section to include alternative interpretations

*Suggested changes:*

*1) In E. coli, the two replisomes from two forks overlap with each other for a significant portion of time (such as in Figure 1—figure supplement 2 panel C. 30min to 40 min to 50 min). All analyses are done under the assumption that a large fluorescence spot could contain either one or two replisomes (cannot be resolved visually due to short distance separation). Perhaps the authors should state this more explicitly in the text and analyses.*

We have included the following sentence in the Methods section:

“We expect that co-localization of sister replisome will have no effect on the rates calculated since the intensity of every spot is normalized against itself in FRAP, and the average rate of recovery is the same at every replisome. Similarly, in sptPALM binding time of individual molecules should not be influenced by a nearby replisome, resulting only a minimal increase in the probability of re-binding to the same place”.

*2) In discussing the FRAP results for DnaB, the authors attribute the incomplete fluorescence recovery to slow diffusion of DnaB. This logic isn't clear and should be explained in greater detail.*

We agree that the original explanation was misleading. We have now removed the speculation on the diffusion rate of DnaB and simplified the description of the recovery curve for DnaB.

While doing imaging, we observed that the fluorescence of the diffusing pool of DnaB was not as evenly distributed as for other subunits, often appearing similar to the spots formed by the immobile copies. This qualitative observation fits with a slower diffusion rate estimated using PALM (Figure 2—figure supplement 4). We speculated that the spot-like fluorescence distribution of the diffusing molecules could complicate accurate intensity estimation by the tracking algorithm. This was the reason we included it in the Results section.

However, we recognize that discussing diffusion rates at that point may confuse the reader. Besides, it is unnecessary since even in the diffusion-uncoupled scenario the FRAP curve is expected to be formed by two separate segments: an initial spike in recovery lasting for a short a duration due to diffusion, followed by a slower recovery rate representing unbinding (McNally JG, Methods Cell Biol, 2008, 85:329). The initial spike in the recovery curve of DnaB is entirely consistent with the diffusing population moving into the bleached area.

*3) Figure 1 and Figure 4 have three y-axes, and are difficult to comprehend. Separating all the measurements, or using broken bars will be easier for the readers.*

Figure 1 and Figure 4 have been modified.

*4) The data plotted in Figure 2 might be better represented in a table. The bar graph doesn't add anything and could be slightly confusing because the y axes are scaled differently.*

A table has been added.

*5) The authors mentioned using a reaction-diffusion model to analyze FRAP traces. There was no description anywhere in the manuscript about what the model is. It appears that the authors simply used an exponential fitting and extracted the half time. If so please state clearly and do not use the name of reaction-diffusion, which has a specific meaning.*

The reaction-diffusion model in a reaction-limited regime does indeed follow an exponential behavior. We have modified the Methods section to make this clearer.

*6) There are typos in Figure 2 and Figure 2—figure supplement 3 ("Bleach time" legend).*

Tbleach was replaced by Bleach time in the figures.

*7) Please provide n values (sample size) for all statistic measurements with error bars.*

Please refer to [Supplementary-material SD1-data].

*8) Regarding additional data files and statistical comments – the authors should provide n values for all measurements.*

Please refer to [Supplementary-material SD1-data]

[Editors' note: further revisions were requested prior to acceptance, as described below.]

*Primary points:*

*1) Some concerns about the quality of the lifetime fits for the PALM data still exist. From the presentation of the data in the figures, it's very hard to assess the quality of the fits, and many of the conclusions of this paper (exchange of the entire Pol III holoenzyme, no evidence of distinct leading and lagging-strand polymerase dynamics, etc.) rely heavily on those lifetimes. The response raised in the previous version about the presentation of data in histograms and the number of bins used is not wholly clear. For example, the epsilon lifetime data in Figure 2—figure supplement 3 appear to include 143 molecules for the 500 ms exposure and 415 molecules for the 2s exposure. Yet the histograms appear to contain data in only 5 or 6 bins, which is not consistent with the square root of N binning claimed to be used. Even if the MLE fitting method is insensitive to the choice of bins, the current presentation of the data makes it difficult to see how well the data are fit by the resulting curves. Please resolve this issue.*

We have now modified Figure 2 and Figure 2—figure supplement 3. In both cases, we have removed binning from the plots of ε subunit to facilitate the assessment of the fits to our data. We have maintained binning for DnaB in Figure 2 since the sample size is smaller and the data is distributed over wider range.

We have also added [Supplementary-material SD1-data] to present the results of Chi-square test, used to assess the goodness-of-fit in our data. In this same table, we also describe the results of two other tests used to examine the likelihood of a two-exponential behavior in our data or, in other words, test if the single-exponential model represents the best model compared to possible alternatives. The results support fitting with a single-exponential. We have updated the Methods section to describe the application of these tests.

We hope that these changes will make a compelling case for our fitted data.

*2) Regardless of the exact timescales of exchange, it has not been shown that the timescales of the holoenzyme components are identical. The Discussion should be amended to discuss different scenarios for replisome dynamics, such as the possibility that individual components of the holoenzyme exchange relatively rapidly with free proteins in solution without the complex falling apart. The manuscript attempts to argue against this option by noting that epsilon dissociation and rebinding is observed more frequently than one would anticipate from the free pool of epsilon. This argument is tenuous, however, as no direct evidence is provided to show that these epsilon molecules are diffusing with the holoenzyme (to test this, for example, one could determine whether those epsilon molecules that dissociate and rebind have an anomalously slow diffusion constant consistent with the much larger holoenzyme). Please amend the discussion of the model to point out that the favored model is at best suggested by the data, and please also present some alternative interpretations.*

We have now modified the text to account for an alternative explanation where binding of individual subunits is independent of others.

In parallel we have also strengthened our interpretation of turnover as a subassembly of the replisome. First, we point that, as mentioned above, we did not find evidence for a second regime of binding for any of the subunits. In addition, we also highlight that there is only one copy of δ subunit in the clamp loader but it has the same bound time as τ, which is present as three copies. If each copy of τ were to bind independently, we should have expected to observe a longer bound time for δ. This, we think, is the most convincing evidence for our model.

We would like to comment on the statement above where you wrote that: “…it has not been shown that the timescales of the holoenzyme components are identical”. This seems to neglect that our data demonstrates that the subunit bound times are within error of one another. Even though there is still some uncertainty, as it is always the case in research, it is very likely that these subunits share the same time constant.

*Secondary comments:*

*1) Regarding the possibility of two timescales in the lifetime data, it would help the reader assess the quality of these fits by plotting the data on a semi-log axis, so that deviations from linearity are more apparent.*

We have added semi-log plots in Figure 2—figure supplement 3. As mentioned above, we have also added the results of goodness-of-fit tests in [Supplementary-material SD1-data].

*2) Concerning the section "DnaB is a stable platform upon which the PolIII holoenzyme exchanges": Measurement of DnaB's long bound lifetime demonstrates that it is a stable component of the replisome, but not necessarily a platform upon which PolII assembles; this latter claim is likely based on what is known about the replisome, by not experimentally tested here (e.g., a DnaB mutant that fails to interact with tau would need to be used to examine this hypothesis). The section title could be altered slightly to more accurately reflect the experimental results.*

The title now reads: “DnaB may act as a stable platform upon which the PolIII holoenzyme exchanges”. We have also modified slightly the last paragraph of the Introduction section and the title of Figure 3.

*3) Figure 2—figure supplement 4. The color of DnaB and epsilon appears to be switched. In panels E and F, there appeared to be negative bins of the apparent diffusion coefficients? Please fix.*

Figure 2—figure supplement 4 is correct. It shows a clearer peak representing immobile molecules for ε. While for DnaB this peak is not as evident and is difficult to distinguish from the peak of diffusing molecules.

We have corrected Figure 2—figure supplement 4.